# Simple Statistical Models for Predicting Overpressure Due to CO$_2$ and Low-Salinity Waste-Fluid Injection into Deep Saline Formations

Esmail Ansari [1],*, Eugene Holubnyak [2] and Franciszek Hasiuk [1]

[1] Kansas Geological Survey, University of Kansas, Lawrence, KS 66047, USA
[2] School of Energy Resources, University of Wyoming, Laramie, WY 82071, USA
* Correspondence: esmailansari1@gmail.com

**Abstract:** Deep saline aquifers have been used for waste-fluid disposal for decades and are the proposed targets for large-scale CO$_2$ storage to mitigate CO$_2$ concentration in the atmosphere. Due to relatively limited experience with CO$_2$ injection in deep saline formations and given that the injection targets for CO$_2$ sometimes are the same as waste-fluid disposal formations, it could be beneficial to model and compare both practices and learn from the waste-fluid disposal industry. In this paper, we model CO$_2$ injection in the Patterson Field, which has been proposed as a site for storage of 50 Mt of industrial CO$_2$ over 25 years. We propose general models that quickly screen the reservoir properties and calculate pressure changes near and far from the injection wellbore, accounting for variable reservoir properties. The reservoir properties we investigated were rock compressibility, injection rate, vertical-to-horizontal permeability ratio, average reservoir permeability and porosity, reservoir temperature and pressure, and the injectant total dissolved solids (TDS) in cases of waste-fluid injection. We used experimental design to select and perform simulation runs, performed a sensitivity analysis to identify the important variables on pressure build-up, and then fit a regression model to the simulation runs to obtain simple proxy models for changes in average reservoir pressure and bottomhole pressure. The CO$_2$ injection created more pressure compared to saline waste-fluids, when similar mass was injected. However, we found a more significant pressure buildup at the caprock-reservoir interface and lower pressure buildup at the bottom of the reservoir when injecting CO$_2$ compared with waste-fluid injection.

**Keywords:** CO$_2$ sequestration; waste-fluid injection design; carbon capture utilization and storage; CCUS; US midcontinent; Kansas

## 1. Introduction

World energy needs are expected to accelerate carbon emissions and increase CO$_2$ atmospheric concentrations over the next decades [1]. Carbon Capture, Utilization and Storage (CCUS) is a basket of technologies that could help to mitigate global CO$_2$ emissions and induce economic development. For CCUS to contribute to reducing the accumulation of CO$_2$ in the atmosphere, it must operate at a large scale, one comparable to global oil production [2,3]. In the USA, the recently expanded U.S.A. Internal Revenue Service Section 45Q tax credits (45Q) and multiple other initiatives have resulted in a renewed interest in CCUS technology [4,5].

Deep saline aquifers (>800 m) are important candidates for CO$_2$ disposal because they are common and, if characterized properly, offer low risks associated with injection and storage. One risk that can exist with this class of reservoirs is when they overlie crystalline basement rocks that contain faults at critical equilibrium [6]. Seismicity can be induced if waste fluids cause enough of a pressure change to upset these basement faults. This has occurred in Texas [7], Oklahoma [8], and to a lesser extent in Kansas [9]. To address concerns with waste-fluid injection, it is necessary to model the effect of reservoir pressure increase on

$CO_2$ movement into geological layers [1]. Pressure changes due to waste-fluid injection also could limit storage capacity and injectivity by causing pressure interference and overlap when multiple injection wells are operating in close proximity [10]. Three key reservoir metrics for the success of a waste-fluid injection project are: (1) injection capacity for each well (injectivity), (2) reservoir storage efficiency/capacity, and (3) long-term containment (i.e., ensuring injected fluid stays within the storage formation). All of these properties are influenced by pressure change in the reservoir [11–13]. Therefore, a comprehensive understanding of pressure buildup in and propagation through the reservoir are keys to the success of any waste-fluid injection project.

For this study, we used synthetic yet representative models for the Patterson Field, which is the proposed target for injection and storage of 50 Mt of $CO_2$ over 25 years. Patterson Field is located in western Kansas and consists of three saline formations, including the Arbuckle Group aquifer, which is the target injection zone for the current study. The Cambro-Ordovician Arbuckle aquifer is a giant heterogeneous and fractured dolomitic aquifer spanning several states across the Southern USA Midcontinent region [14]. The Arbuckle is the principal disposal zone in Kansas and has been used for waste-fluid disposal for several decades. Currently, thousands of wells inject hundreds of millions of tons of waste-fluid into this aquifer [9]. The aquifer is also favorably located between multiple major sources of $CO_2$ emissions, existing pipeline systems, and proposed pipeline infrastructures to the major storage and EOR sites [15]. A detailed study of 10 sites in Kansas showed that these sites have the potential to store 780 million metric tons of supercritical $CO_2$ in the Arbuckle aquifer alone [16]. If the need for $CO_2$ storage grows, interest in the Arbuckle aquifer would likely expand beyond the 10 sites already identified. However, detailed geological and numerical modeling of $CO_2$ injection into each Arbuckle area is time consuming and computationally expensive when applied on a case-by-case basis. In addition, many reservoir parameters, such as porosity and permeability, have uncertainty during simulation due to the geological heterogeneity of the Arbuckle. For these reasons, researchers often favor analytical solutions to understand the generic underlying physics of a system before proceeding to detailed case-by-case simulation [17,18]. The problem with analytical solutions, however, is that the inherent assumptions made in simplifying the system can also lead to debate as to their applicability [19]. An alternative approach is to run numerical simulations and translate the results into generic "proxy models" (also known as reduced-physics models or predictive models), which represent the original simulation results. In this approach, a combination of numerical simulations and statistical methods is used to develop models that can quickly screen and select appropriate sites for $CO_2$ injection [20].

Simplified-physics proxy modeling has been used previously in the context of $CO_2$ injection into saline aquifers [21,22]. Anbar and Akin [22] used Latin-Hypercube sampling and regression models to develop predictive models for quick estimation of $CO_2$ storage capacity in deep saline aquifers [22]. They observed that linear least-square models performed better than their nonlinear counterpart. Schuetter et al. [21] compared different types of proxy models, including polynomial regression, kriging, MARS (multivariate adaptive regression spline), AVAS (additivity and variance stabilization), and TPS (thin plate spline) [21]. They performed k-fold cross validation on the developed models and concluded that for predicting cumulative volume of $CO_2$ injected and maximum $CO_2$ flux entering the caprock, the kriging formulation yielded the most favorable root mean squared error (RMSE). Schuetter and Mishra [23] compared several sampling techniques for modeling plume radius and total storage efficiency, including Box—Behnken and Latin-Hypercube, and found that the Box—Behnken sampling method performed the best for sampling the parameter space [23]. They also found that the kriging formulation provided only marginal advantage over the quadratic regression formulation using LASSO (least absolute shrinkage and selection operator) and recommended the latter methods because of efficiency and simplicity. Ganesh and Mishra [24] developed simplified-physics proxy models for $CO_2$ storage and plume extent as a function of reservoir and caprock properties [24].

They considered confined heterogeneous reservoirs for their modeling and assumed each well ultimately established a closed drainage area. They conclude that their simplified physics-based model can reasonably predict storage efficiency ($E_s$) and could be used as an alternative for rapid feasibility, system design and permitting $CO_2$ sequestration sites.

The objective of this work is to develop and validate generic proxy models for calculating bottomhole and average reservoir pressure buildup for $CO_2$ injection and low-salinity water injection into deep saline aquifers. Because of limited experience with $CO_2$ injection and similarities between $CO_2$ injection and waste-fluid disposal practices, this work compares these two injection practices. This paper proceeds as follows: First, we provide background information about injection in Kansas. Data from the Patterson site, located in western Kansas, are used to create numerical models with a range of uncertainty in the model parameters. We run full-physics numerical simulation selected by an experimental design scheme and perform a sensitivity analysis to identify important properties that influence the response (i.e., pressure build-up). We then create proxy regression models using the important features and use the models to perform a Monte Carlo uncertainty analysis simulation. Our study showed that when injecting $CO_2$, the pressure buildup was higher at the top of the aquifer compared with the buildup associated with injecting a similar mass of low-salinity water. The presented statistical models can be used as an early screening tool to screen large databases of reservoirs to select the most attractive ones for $CO_2$ storage. These models are especially useful when geological data on the reservoir are limited (e.g., deep saline aquifers across the US midcontinent).

## 2. Background

### 2.1. Injection in Kansas

The Environmental Protection Agency (EPA) regulates several classes of waste fluid injection wells in the USA unless states have been granted primacy over some (or all) [25] of these classes. Classes I, II, and VI wells are of relevance to this study. Class I wells inject municipal or industrial waste fluids. Class II wells inject petroleum field waste fluids or conduct enhanced oil recovery (EOR) operations. Class VI wells inject $CO_2$ for deep saline storage or EOR.

Every year, more than 2400 injection Class II and 49 Class I wells across Kansas dispose of more waste-fluid into the Arbuckle formation alone (~120 Mt/year) than a commercial mass of $CO_2$ (~2 Mt/year) that may be injected into the Arbuckle, Viola, and Osage Formations [9]. A commercial $CO_2$ project requires injecting 50 Mt over 25 years (or ~2 Mt/year) into these formations to obtain reasonable injectivity and minimize the pressure buildup around each injection well. Assuming the Arbuckle would accept 50% of the total injection using six Class VI injection wells, the injected volume would be 0.17 Mt/year into the Arbuckle in each well. In 2016, 135 injection wells (20 Class I wells and 114 Class II wells) across Kansas exceeded this rate (Figure 1). The Arbuckle formation at the Patterson site has a temperature of 50 °C and a pressure of 11.8 MPa, a condition in which $CO_2$ density is $\approx 500$ kg/m$^3$, half that of water (assuming water density of 1000 kg/m$^3$). Pressure change is a function of volume change and a volume-equivalent amount of water for the 0.17 Mt/year $CO_2$ would be 0.34 Mt/year. Currently, 42 wells (16 Class I wells and 26 Class II wells) across Kansas inject more than 0.34 Mt/year (Figure 2). Because the Arbuckle aquifer is hydraulically connected to the critically stressed basement [8], one challenge is to determine the rate at which $CO_2$ could be injected without resulting in a pressure buildup that could initiate seismicity on preexisting faults. Thus, developing statistical proxy models to estimate pressure buildup in the well and the reservoir is essential.

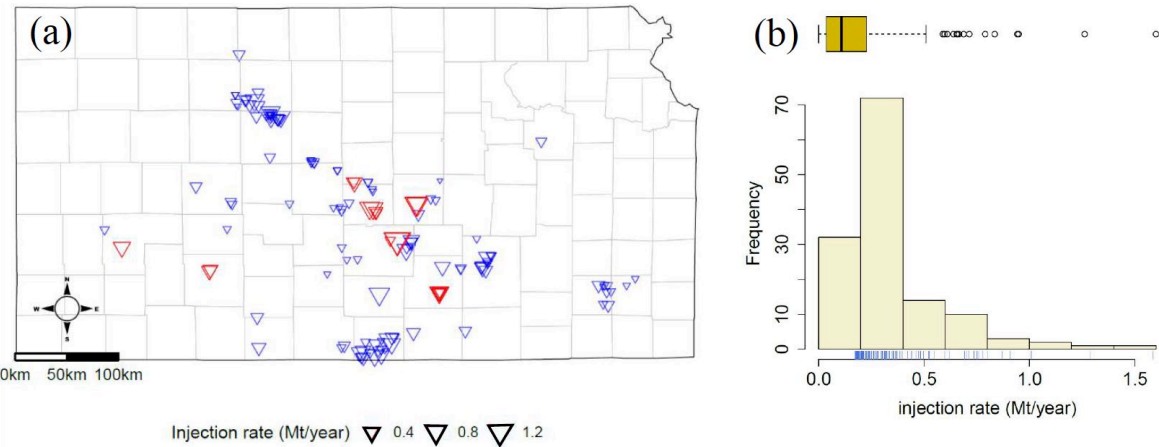

**Figure 1.** (**a**) Wells injecting more than 0.17 Mt/year of waste fluid into the Arbuckle aquifer across Kansas. The red triangles show Class I and the blue triangles show Class II injection wells. (**b**) Histogram of injection rates for the wells injecting >0.17 Mt/years waste-fluid across Kansas (circles indicate outliers).

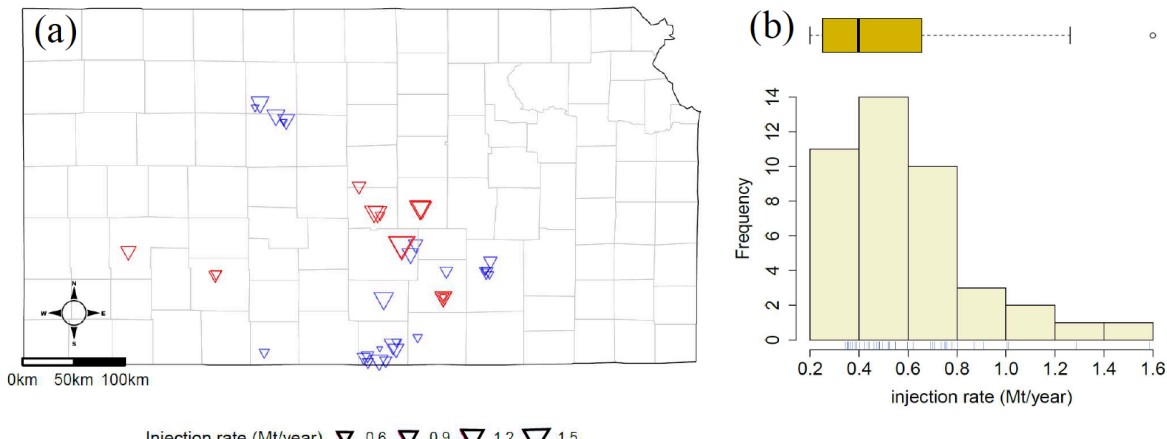

**Figure 2.** (**a**) Wells injecting more than 0.34 Mt/year of waste fluid into the Arbuckle aquifer across Kansas. The red triangles show Class I and the blue triangles show Class II injection wells. (**b**) Histogram of injection rates for the wells injecting >0.34 Mt/years waste-fluid across Kansas (circles indicate outliers).

*2.2. Western Kansas*

Figure 3a shows the Patterson area (Kearny County, western Kansas) wells that inject into the Arbuckle aquifer and their injection volumes, including the City of Lakin well (KS-05-093-002) for which fall-off test measurements are available [26]. The right figures show the current injection volume in the Patterson area. Figure 3b shows the injection volumes into the Arbuckle aquifer around KS-05-093-002. The rate of injection within 25 km of the well is less than 0.1 Mt/year (the $CO_2$ project roughly requires 0.17 Mt/year per well into the Arbuckle and 1 Mt/year into all six wells, which are less than 25 km apart). The total injection into this aquifer from 2000 to 2017 (17 years) within 35 km of the well amounted to less than 4.5 Mt (a $CO_2$ storage project would require ~17 Mt injection into the Arbuckle over 17 years). It is advantageous to inject $CO_2$ in western Kansas, specifically at the Patterson site (Figure 3b), because it has low historical and currently active injection volumes. One uncertainty for the Patterson site, however, is how injecting 50 Mt of mass would affect the pressure of the Arbuckle aquifer, when it would represent an order of magnitude higher injection rate and cumulative injection than all previous injections. Reservoir simulation is required to address this uncertainty. In this

study we use a statistical proxy model to predict the pressure increase at the injection well and throughout the reservoir for low-certainty model parameters and under different injection scenarios.

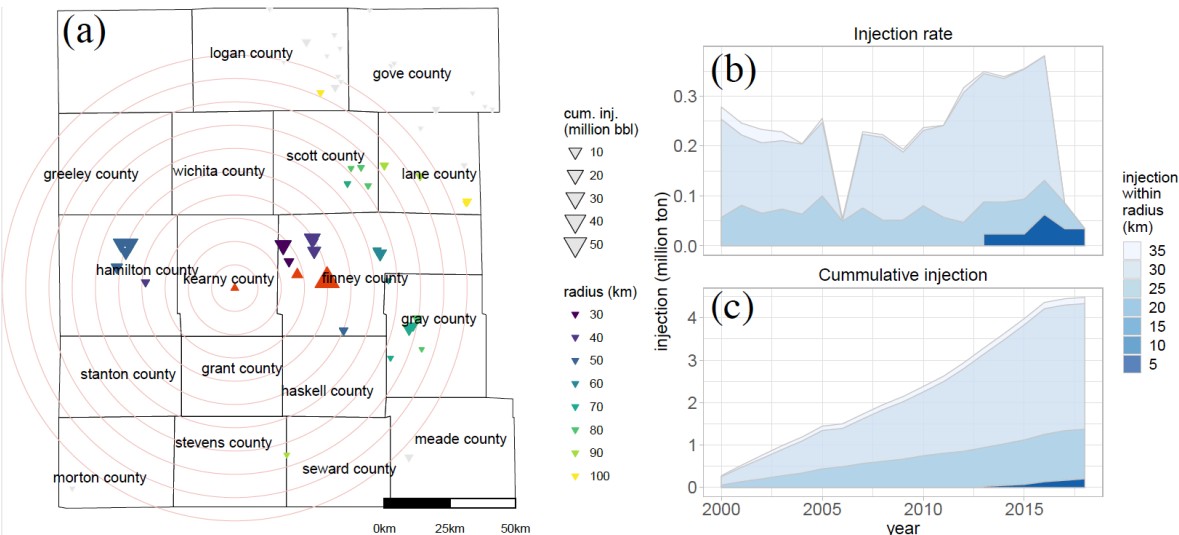

**Figure 3.** Patterson area (**a**) wells that inject into the Arbuckle aquifer in western Kansas and their injection volumes, including the City of Lakin well for which fall-off test measurements are available (circles show 10 km increments). Red upward triangles represent Class I wells for which more data are available, and downward triangles represent Class II injection wells. Injection rate (**b**) and cumulative injection (**c**) volume in the Patterson area are also shown.

## 3. Methods

The strategy in this study was to use typical geological and reservoir properties data for the Arbuckle aquifer in the Patterson area in western Kansas to create representative numerical reservoir models for the area. A numerical compositional simulator package, GEM (CMG Ltd., Canada, cmgl.ca/gem), was used for simulations. Different dynamic models were built by sampling the parameter space. We considered two similar mass injection scenarios: (1) $CO_2$ injection and (2) low-salinity waste-fluid injection. Two responses are investigated: (1) bottomhole pressure increase ($\Delta p$) in the well after 25 years of injection and (2) average pressure increase in the reservoir (averaged over the entire reservoir volume) after 25 years of injection. The reservoir is connected to an open analytical aquifer on each side and to a low permeability/low porosity analytical aquifer at the bottom (representing the Precambrian basement). The reservoir is sealed from top.

### 3.1. Workflow for Creating Models

The workflow for creating the models consisted of the following steps: (A) sample the parameter space using experimental design, (B) run the full simulation using the sampled parameter values, (C) perform the sensitivity analysis to distinguish important parameters, (D) translate the simulations into proxy models using regression analysis, and (E) use the proxy models instead of simulation to perform Monte Carlo simulation on the probability distributions assigned to the parameters (all parameter distributions are assumed uniform). A brief description of each step follows.

### 3.2. Base Reservoir Model

The average reservoir permeability is expected to be a known reservoir parameter obtained from core measurements or well tests (DST or fall-off tests). For this study, reservoir-scale permeability of the Arbuckle was estimated to be 500 mD based on fall-off test results [26]. For creating generic statistical models, simple Cartesian grid blocks (15 × 15 × 25) with incremental distances of 200 m × 200 m × 10 m were used for the

base case. $CO_2$ or low-salinity water was injected at a similar mass rate of 0.17 Mt/year for 25 years, after which the well is shut in. Because experimental data on drainage and imbibition processes were unavailable for the Patterson Field, relative permeability and capillary pressure curves [27] (Figure 4) and reservoir properties (Table 1) obtained from other field studies on the Arbuckle aquifer were used in the modeling. Salinity was used only in the waste-fluid injection modeling. Rowe-Chau and Kestin correlations were used to calculate water density and water viscosity from salinity values, respectively [28]. To simplify results, this study did not consider temperature gradient as a parameter because studies have shown that the cooling front extends only a few hundred of meters from the injection well after decades of injection [29].

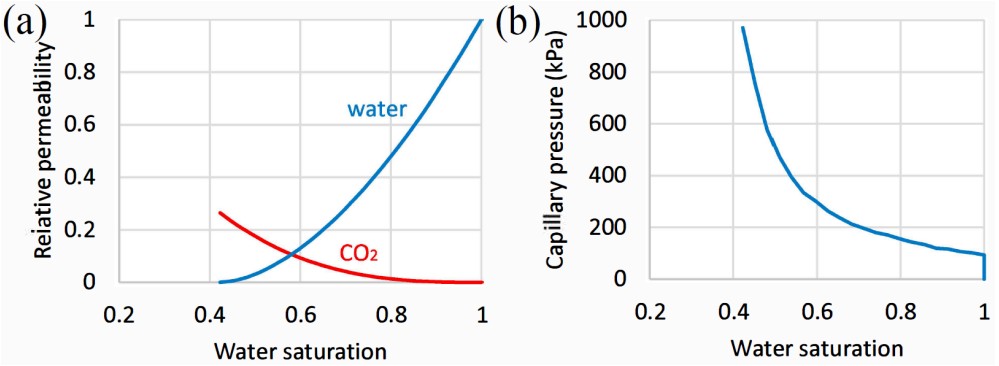

**Figure 4.** Relative permeability (**a**) and capillary pressure (**b**) curves for the Arbuckle aquifer [27].

**Table 1.** Properties used in the numerical simulation.

| Parameter | Minimum | Base Case | Maximum |
|---|---|---|---|
| Rock compressibility (1/kPa) | $1 \times 10^{-8}$ | $1 \times 10^{-8}$ | $1 \times 10^{-7}$ |
| Injection rate ($m^3$/day) | 300 | 450 | 600 |
| $K_z/K_x$ | 0.05 | 0.1 | 1 |
| Permeability (mD) | 200 | 500 | 700 |
| Porosity | 0.05 | 0.1 | 0.15 |
| Temperature (°C) | 30 | 50 | 60 |
| Pressure (kPa) | 12,000 | 15,000 | 18,000 |
| Salinity (mg/L) | 1000 | 5000 | 30,000 |

Relative permeability and capillary pressure data were based on a field study of the Arbuckle aquifer in the Wellington Field in south-central Kansas and the Cutter Field in southwestern Kansas, for both of which KGS hosts continuous coring, modern wireline well log data including nuclear magnetic resonance (NMR), and well test data [30]. Continuous core data and NMR for Wellington-KGS-1-32 were used to derive generalized capillary pressure curves and irreducible water saturations. Based on the estimated end points and data from the Cutter-KGS-1 well, nine sets of relative permeability and capillary pressure curves for both drainage and imbibition were generated for the Arbuckle reservoir using a formula that correlates them to the end points and the reservoir quality index (RQI) [27].

### 3.3. Boundary Condition

We assumed open boundary conditions because the Arbuckle aquifer is a regional, hydraulically connected carbonate aquifer, which can be assumed to be an infinite-acting aquifer. The reservoir model is connected to an analytical Carter—Tracy aquifer in which leakage is allowed [31]. The boundary aquifer has the same properties as the reservoir model. The reservoir model is also connected to an analytical Carter—Tracy aquifer with

low porosity (1%) and permeability (1 mD) at the bottom, which represents the Arbuckle connection to the crystalline basement. The flow leaks from the Arbuckle into the analytical aquifers when the pressure is high in the reservoir and due to the hydrostatic pressure gradient between the reservoir and the basement rock, which is caused by fluid injection.

*3.4. Initial Condition*

The initial condition constitutes a pressure of 11.8 MPa at a depth of 1800 m and isothermal condition with uniform temperature of 50 °C.

*3.5. Defining the Response*

The study defines response as the pressure buildup around the wellbore (i.e., $\Delta$BHP) and average pressure buildup in the reservoir ($\Delta P_{avg}$) at the end of the injection period (25 years).

*3.6. Box—Behnken Design*

Two response surface methods are frequently used for developing second-degree quadratic models: central composite design and Box—Behnken design (Figure 5) [32,33]. These methods differ in the sampled parameters for a three-dimensional parameter space, in which the corners of the cubes show the maximum and minimum values for the parameters. The central composite design selects the corners while the Box—Behnken design selects parameter averages. We investigated seven factors (i.e., model parameters), listed in Table 1, and used a central composite design because although the central composite design (CCD) requires more runs, it is more accurate at the simulated points. The CCD experimental design required 81 training and 4 testing runs (total of 85 runs) for these seven parameters used. The sampled models were run in batches of 25 runs and each batch took ~8 min to complete.

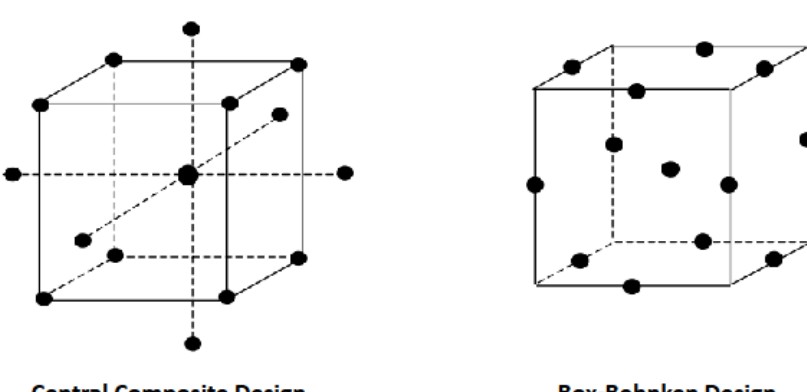

**Central Composite Design**　　　　　　　　　　**Box-Behnken Design**

**Figure 5.** Two types of experimental designs.

*3.7. Sensitivity Analysis*

This study employed two methods to determine which input parameters have the greatest effect on reservoir response and to discern important factors: (1) Sobol analysis [34] and (2) Tornado charts.

*3.8. Regression Model*

A simple quadratic regression equation was fitted (Equation (1)) for proxy modeling the simulation results:

$$y = a_0 + \sum_{j=1}^{k} a_j x_j + \sum_{j=1}^{k} a_{jj} x_j^2 \tag{1}$$

### 3.9. Model Reduction and Testing

After the regression models were created, the study used a model-reducing algorithm (Best Subset Selection [35]) to determine the best reduced model. A reduced model has a minimum number of terms. Final models were evaluated using testing runs.

### 3.10. Uncertainty Quantification

Each parameter was assigned a uniform distribution in the range of its minimum and maximum values. Then, a Monte Carlo approach was used to sample the parameter space. The sampled parameters were used in the reduced proxy model to calculate uncertainty in the response.

## 4. Results

### 4.1. Base Case and Response

The pressure increase was simulated around the wellbore and in the reservoir for the base $CO_2$ injection and low-salinity water injection cases under equivalent mass rate ($\approx$1.7 Mt/year as is planned for the Patterson site) after 5, 15, and 25 years of injection (Figure 6). Unlike low-salinity water, in which the pressure increases bottom-up, when injecting $CO_2$ the pressure increased top-down (Figure 6). There are three reasons for this phenomenon. First, hydrostatic pressure is lower at the top of the reservoir and $CO_2$ encounters less resistance to flowing into the reservoir, and second, $CO_2$ has lower density at lower pressures which exist at the top of the reservoir, and third, buoyancy forces keep $CO_2$ flowing at the top of the reservoir. In contrast, low-salinity water has higher density than both reservoir fluids and $CO_2$ so it flowed downward as soon as it was injected into the formation. The maximum pressure buildup for $CO_2$ injection was at the topmost layer, while the maximum pressure buildup for low-salinity water injection occurred at the bottommost layer, where the reservoir is connected to the analytical aquifer at the bottom.

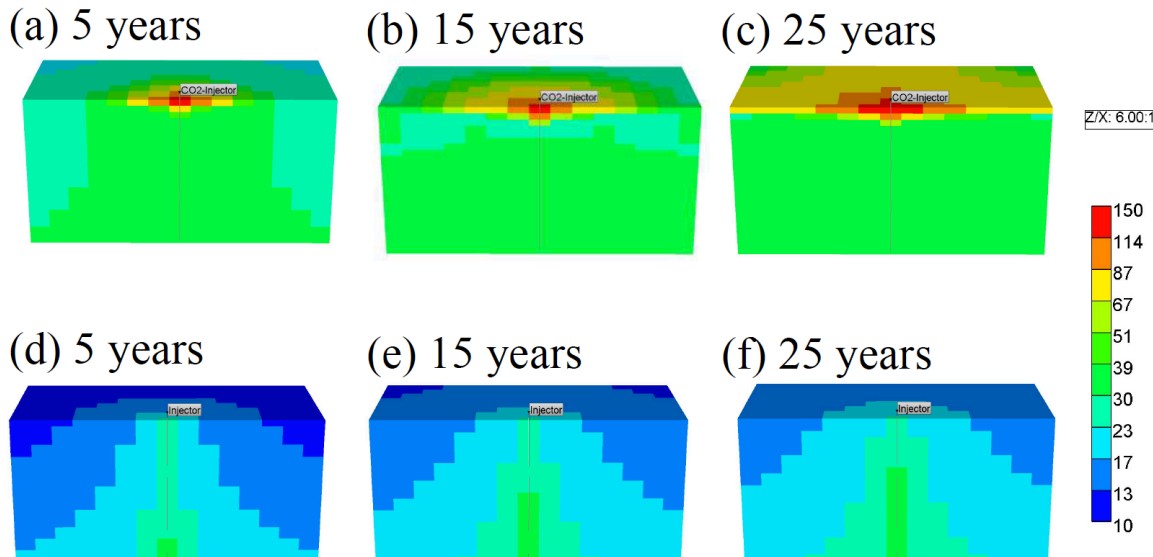

**Figure 6.** Base case simulations showing the pressure plume evolution around the wellbore and in the reservoir after 5, 15, and 25 years for equivalent mass injection of (**a**–**c**) $CO_2$ and (**d**–**f**) low-salinity water.

Figure 7 shows the defined response (pressure buildup in the wellbore and the reservoir) for the base case for supercritical $CO_2$ injection (Figure 7a) and water injection (Figure 7b). For the base case, the maximum pressure buildup in the reservoir when injecting $CO_2$ was $\approx$120 kPa, while the maximum pressure buildup when injecting low-salinity water was $\approx$35 kPa. For both cases, the pressure showed a quick initial increase followed by a subsequent decrease in pressure rate (gradient in time). Because $CO_2$ has

lower density, its corresponding wellbore pressure buildup was higher. The pressure in both wellbore and reservoir for the $CO_2$ injection case quickly declined in injection well pressure. During water injection, the injection well pressure increased monotonically compared with the $CO_2$ injection because of small difference between viscosities, densities, and compressibility of resident brine and injected water.

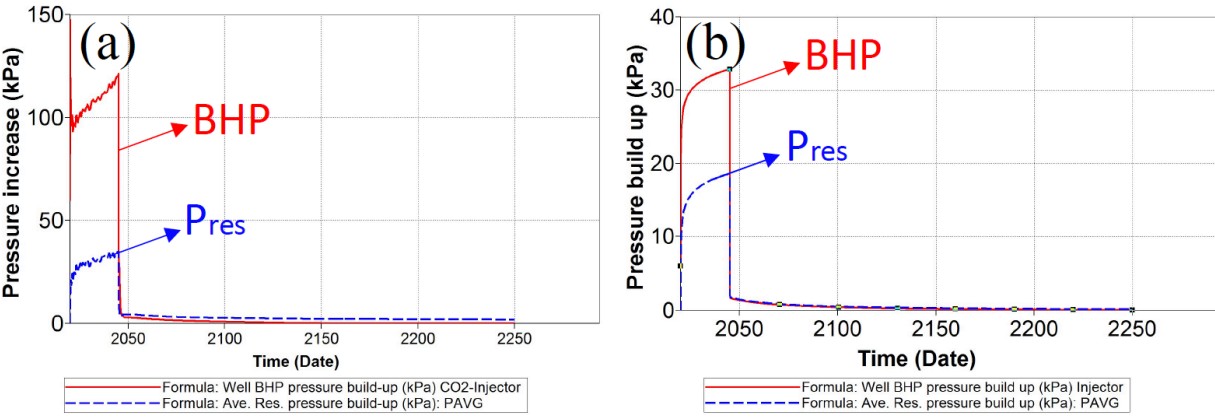

**Figure 7.** Base case simulation showing the pressure buildup in the bottomhole and the reservoir for (**a**) $CO_2$ injection and (**b**) low-salinity water injection.

### 4.2. Sensitivity Analysis

The Sobol sensitivity analysis (Figure 8) was used to estimate the contribution of each factor in predicting the pressure buildup at bottomhole (a) and average reservoir-wide (b) for $CO_2$ injection and similarly for water injection (c,d). Permeability has the highest control on the pressure buildup for both fluid types. When injecting $CO_2$, porosity is the most influential property on reservoir pressure buildup and the second-most influential factor on pressure buildup near the wellbore, but appears to be less influential when injecting water. The effects of reservoir pressure and permeability ratio on $CO_2$ injection are small but they cannot be ignored for modeling because they constitute more than 1% contribution using the Sobol analysis. The effect of rock compressibility in both cases is insignificant. Reservoir temperature affects bottom hole pressure increase when injecting water, but not $CO_2$ (Figure 8b,d). However, the overall reservoir pressure increase after 25 years of $CO_2$ injection mostly depends on initial reservoir pressure (Figure 8b). As expected, the injection rate is important for measuring pressure buildup in all cases.

Tornado sensitivity charts (Figure 9) can be used to detail the effect of important parameters (chosen using the Sobol method, Figure 8) on pressure buildup. For the $CO_2$ injection case (Figure 9a,b), the increase in permeability and porosity decreases BHP and average reservoir pressure, respectively. An increase in temperature decreases BHP because it reduces $CO_2$ viscosity. In addition, an increase in pressure decreases pressure buildup in the reservoir because $CO_2$ has higher density at higher pressures, requiring less storage volume. Although many factors are involved in $CO_2$ storage, ultimately only these important factors are required for proxy modeling. Similarly, for the low-salinity water injection case (Figure 9c,d), an increase in permeability and temperature decreases the pressure buildup in the bottomhole and the reservoir.

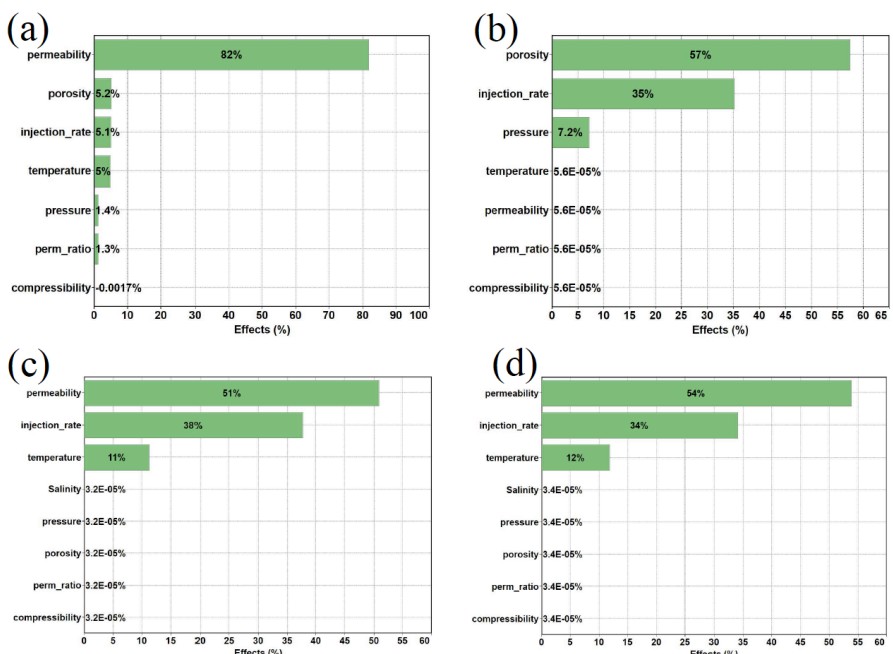

**Figure 8.** Sensitivity analysis using the Sobol method to estimate the percentage each factor contributes to predicting pressure buildup: (**a**) bottomhole pressure buildup for supercritical $CO_2$ injection, (**b**) average reservoir pressure buildup for supercritical $CO_2$ injection, (**c**) bottomhole pressure buildup for low-salinity water injection, and (**d**) average reservoir pressure buildup for low-salinity water injection.

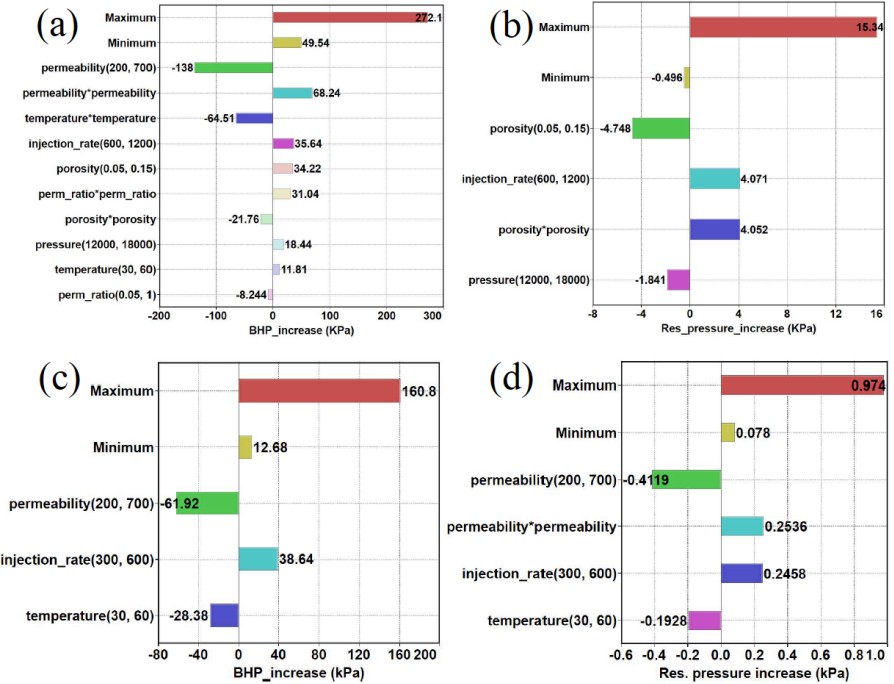

**Figure 9.** Tornado sensitivity charts for the effect of important parameters on pressure buildup: (**a**) bottomhole pressure buildup for supercritical $CO_2$ injection, (**b**) average reservoir pressure buildup for supercritical $CO_2$ injection, (**c**) bottomhole pressure buildup for low-salinity water injection, and (**d**) average reservoir pressure buildup for low-salinity water injection (see Table 1 for the ranges).

### 4.3. Proxy Models

Simulated wellbore pressure buildup can be compared to pressure buildup predicted by the proxy model (Figure 10). Training runs (blue dots in Figure 10) fall almost on the unity line, indicating the model is adequate for reproducing the simulation runs. The four testing runs (green dots in Figure 10) for the models also fall largely within the 95% confidence interval, indicating that the developed models are adequate for predicting simulation runs. In addition, these models for predicting bottomhole pressure buildup perform better than the models developed for predicting average reservoir pressure buildup.

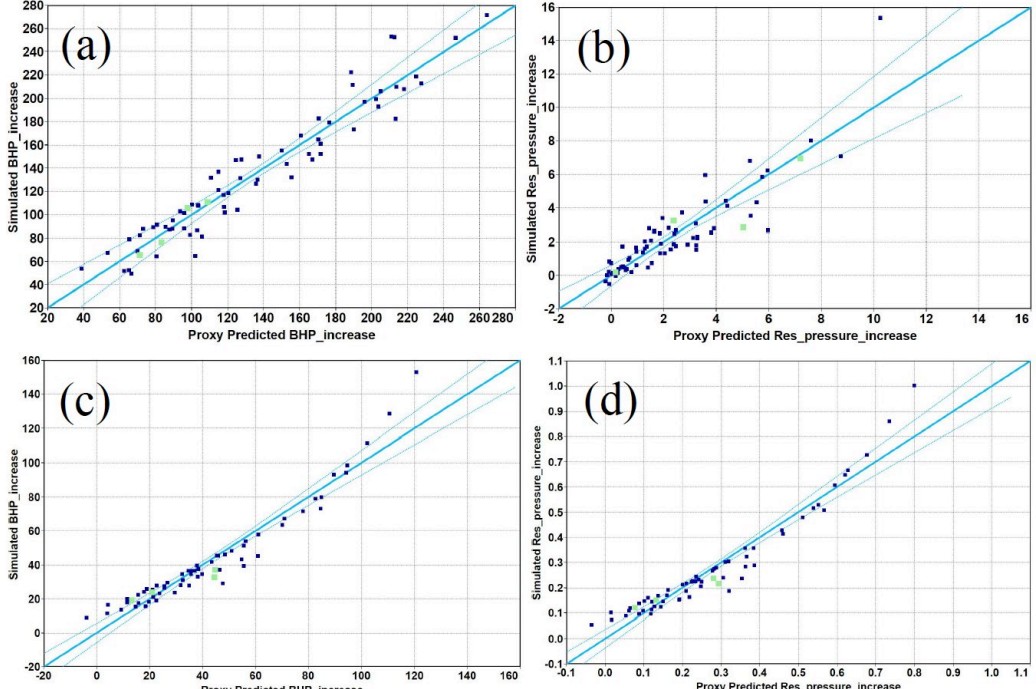

**Figure 10.** Simulated values versus the values predicted by the proxy model. Blue and green dots indicate training and verification runs, respectively, and dotted lines represent the 95% confidence interval around the regression model for (**a**) bottomhole pressure buildup for supercritical $CO_2$ injection, (**b**) average reservoir pressure buildup for supercritical $CO_2$ injection, (**c**) bottomhole pressure buildup for low-salinity water injection, and (**d**) average reservoir pressure buildup for low-salinity water injection. Factors identified as important were used to create the proxy statistical model. A simple quadratic regression was used to connect these parameters to the response.

Equations (2) and (3) are the proxy equations for pressure buildup in the wellbore and reservoir, respectively, due to $CO_2$ injection (units are as indicated in Table 1 and abbreviations are indicated in abbreviation).

$$\Delta BHP = \left(-9.6 \times 10^1\right) + \left(5.9 \times 10^{-2} R_i\right) + \left(-8 \times 10^1 K_{VH}\right) + \left(-7.7 \times 10^{-1} K\right) + \left(1.2 \times 10^3 \phi\right) + \left(1.3 \times 10^1 T\right)$$
$$+ \left(3.1 \times 10^{-3} P\right) + \left(6.8 \times 10^1 K_{VH}^2\right) + \left(5.5 \times 10^1 K^2\right) + \left(4.4 \times 10^3 \phi^2\right) + \left(-1.4 \times 10^1 T^2\right) \tag{2}$$

where $R_i$ = injection rate (m$^3$/day), $K_{VH}$ = permeability ratio (unitless), $K$ = permeability (millidarcies), $F$ = porosity (fraction), $T$ = temperature (°C), and $P$ = pressure (kPa)

$$\Delta P_{avg} = \left(1.3 \times 10^1\right) + \left(6.8 \times 10^{-3} R_i\right) + \left(-2.1 \times 10^2 \phi\right) + \left(-3.1 \times 10^{-4} P\right) + \left(8.1 \times 10^2 \phi^2\right) \tag{3}$$

in which the permeability is in mD, injection rate is in m$^3$/day, temperature is in °C, and pressure is in kPa. Equations (4) and (5) are the proxy equations for pressure buildup in the wellbore and reservoir, respectively, due to low-salinity water injection.

$$\Delta BHP = \left(1.6 \times 10^2\right) + \left(1 \times 10^{-1} R_i\right) + (-4.3\,K) + \left(-9.3 \times 10^{-1} T\right) + \left(3.6 \times 10^{-4} K^2\right) \tag{4}$$

$$\Delta P_{avg} = (1.1) + \left(6.5 \times 10^{-4} R_i\right) - \left(3.0 \times 10^{-3} K\right) + \left(-6.4 \times 10^{-3} T\right) + \left(2.5 \times 10^{-6} K^2\right) \tag{5}$$

### 4.4. Uncertainty Analysis

Because of a lack of a priori knowledge for the distribution of the unknown parameters, we assigned a uniform distribution to all parameters. We used the developed model in a Monte Carlo simulation approach to understand the uncertainty in BHP increase and average reservoir pressure increase after 25 years (Figure 11).

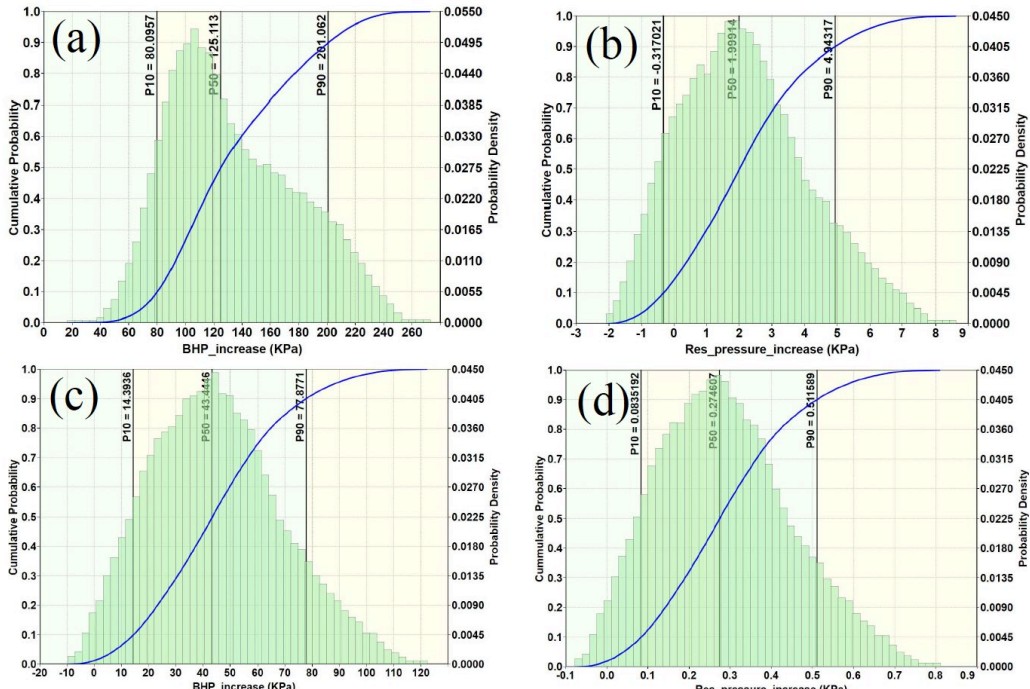

**Figure 11.** Uncertainty in the pressure buildup using 65,000 Monte Carlo simulation runs: (**a**) bottomhole pressure buildup for supercritical $CO_2$ injection, (**b**) average reservoir pressure buildup for supercritical $CO_2$ injection, (**c**) bottomhole pressure buildup for low-salinity water injection, and (**d**) average reservoir pressure buildup for low-salinity water injection.

### 4.5. Comparison with Detailed Geological Modeling and Numerical Simulation Results

Static geological reservoir models were constructed to simulate $CO_2$ injection into each reservoir interval (Osage-Warsaw, Viola, Arbuckle) using numerical simulation. The structural framework was built by using elevation data for top and base of reservoir from 20 wells. Porosity and permeability were modeled away from wellbore petrophysical logs and core analysis data using Kriging. The static geological modeling had an average permeability of ~15 mD obtained from well logs. The static models were then simulated numerically and results show a pressure increase of 4 MPa (600 psi), which mostly occurs near wellbores (Figure 12). The proxy models developed for this study use a median permeability of 500 mD, obtained from fall-off well test measurements in the area, which is 40 times more than the value suggested by well logs. As mentioned, permeability was the most important factor controlling the pressure increase near the wellbore (Figure 8). The proxy models suggest a pressure increase of 0.125 MPa for the Patterson area (Figure 11a), roughly 1/40th of the pressure increase obtained from the static geological model. This ratio corresponds to the permeability ratio of 1/40th, used in the static geological model and indicates consistency between the two modeling approaches.

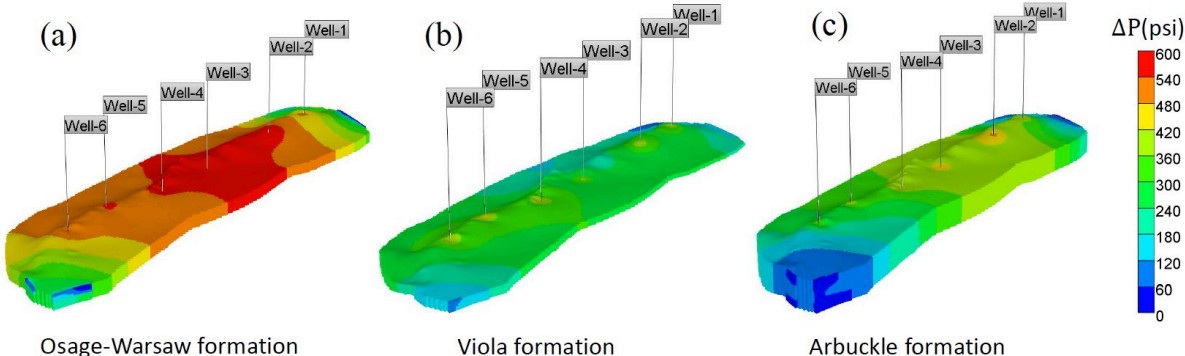

**Figure 12.** Static geological models with simulation results of the Patterson Field after 25 years of injection using porosity and permeability values interpreted from well logs and core analysis data.

## 5. Discussion

The overpressure in the wellbore and pressure plume in the reservoir are expected to diffuse and disappear shortly after injection stops (Figure 7). The hydrological properties of the disposal zones (i.e., porosity, permeability, and injection rate) control the pressure increase and propagation in the subsurface, ultimately determining the long-term fate of the $CO_2$ and associated risks (Figure 8). Porosity and permeability, which are geological factors, and to a lesser extent injection rate, which is an operational factor, are also the factors with the greatest uncertainty in this (or any) project. As porosity or permeability increase, the pressure buildup around the wellbore decreases because the injected fluid can advect away from the wellbore more easily (Figure 9). Because the Arbuckle is a fractured carbonate aquifer, there is a high uncertainty in the distribution of permeability and porosity in the aquifer, especially for "extreme permeability" features like fractures. Porosity, in particular, controls storage capacity and decreases with depth because of compaction [36]. Another important factor modifying the absolute permeability values for $CO_2$ or waste fluid is the relative permeability corresponding to each injection scenario, which was assumed to be fixed in this study (Figure 4). In general, relative permeability curves are uncertain, have heterogeneity for carbonate reservoirs, and are difficult to obtain. In all simulations run, the fluid was injected into the entire wellbore interval exposed to the reservoir. However, because of lower hydrostatic pressure at the top of the aquifer, and $CO_2$'s lower compressibility and density, it flowed more easily into the top of the aquifer until the pressure increase overcame the hydrostatic gradient (Figure 6). This follows reality, where the injectant plume advances from top to bottom of a reservoir and not through the entire injection interval of the wellbore [37].

Interplay between three forces governs $CO_2$-brine flow dynamics: viscous, gravitational, and capillary forces. The proxy models and equations describing the $CO_2$ dynamics near the wellbore (Figure 10a and Equation (1)) captured the viscous and gravitational forces. Temporal variation in near-wellbore pressures depends strongly on the contrast between injected fluid density and resident brine density. Viscous forces grow the $CO_2$ (or water) plume cylindrically, gravitational forces spread it vertically toward the base of the caprock, and capillary forces trap the $CO_2$ in pores. In practice, however, the heterogeneity of both the rock's permeability and capillarity hinder a purely cylindrical $CO_2$ plume, extending the $CO_2$ along higher permeability streaks, zones, or features. Because low-salinity waste fluid is denser and more viscous than $CO_2$, its injection is dominated by viscous forces while the $CO_2$ injection is dominated more by gravitational forces and gravity override (Figure 6). The domination of viscous forces causes low-salinity waste fluid to advance with a steeper front. If the $CO_2$ and low-salinity water are injected at surface temperature (~18 °C mean annual temperature for the Patterson area) instead of subsurface (50 °C for the Arbuckle reservoir), a slower, cooler low-salinity waste fluid moving across the entire thickness of the aquifer equilibrates more quickly with the formation

temperature than the cooler $CO_2$ plume advancing near the caprock. Reduction in the temperature of the caprock results in increase in effective normal stress due to thermal contraction of the caprock and dampens overpressure. Overpressure at the caprock due to injection can increase pore pressure and induce caprock fracturing. Fracturing occurs when pore pressure exceeds minimum principal stress. The threshold value for the Arbuckle caprock can be measured from geomechanical tests. Formation pressures are highest at the wellbore, making caprock around the wellbore more susceptible to pressure changes, requiring a detailed geomechanical analysis of the caprock formation using wellbore core samples (Figure 6). Both $CO_2$ and low-salinity water can move denser formation fluids into the basement and induce seismicity along basement faults. Pressure buildup of about 0.6 MPa in the Arbuckle has been observed to induce seismicity in the basement [9]. If the permeability of the reservoir is on the order of 500 mD (as suggested by well tests), the chance of induced seismicity is lower (Figure 11). If the permeability is on the order of 10 mD, as is suggested from well logs, seismicity maybe induced in the basement due to pressure buildup in the Arbuckle (Figure 12).

This study used a $k_v/k_h$ ratio to describe the heterogeneity of the reservoir (Table 1, Figure 9a). In homogenous reservoirs, the buoyant $CO_2$ does not flow through a large portion of the rock volume due to gravitational forces. However, in heterogeneous reservoirs, the resulting disperse flow paths can increase the portion of the rock that comes into contact with the flow. Thus, heterogeneity and geometry of depositional facies can control storage efficiency, capacity, and injectivity. The Arbuckle aquifer is highly heterogeneous and is known to have fractures and vugs with permeability ranging from a few milliDarcy to more than one Darcy [14,38]. Thus, the actual $CO_2$ plume development in the reservoir is expected to be strongly controlled by uncertainty in these geological factors. In addition, the Arbuckle contains barriers and baffles (e.g., shales); therefore, it is expected that the developed plume will be multi-layered and counteract the effect of buoyancy, which may cause the plume to quickly rise to the top of the Arbuckle.

This study has several limitations due to limited availability of dynamic data (e.g., formation pressure data) for the Patterson area. These statistical models are relevant when the reservoir parameters are within the ranges of parameters given in Table 1. Proper site characterization is necessary for using proxy models because they require average values for each parameter. Well pressure monitoring is required during and after $CO_2$ injection stage and these data should be used for tuning dynamic modeling. A downhole measurement of pressure and temperature eliminates the uncertainties in calculations based on wellhead pressure and temperature and should be used in Patterson area development. Anomalies in the developed pressure models compared to field pressure measurements can be used to detect the leakage through caprock or seal. Measuring the actual pressure plume extent in the reservoir, however, requires geophysical measurements (e.g., seismic, gravity, and satellite data). One method for detecting the elevated pressure at the Patterson site and inferring the flow within the reservoir can be to measure the elastic deformation of surface rocks using high-quality interferometric synthetic aperture radar (InSAR) observation, which is shown to be successful at other $CO_2$ injection projects such as In-Salah Field, Algeria [39]. Finally, the presented proxy models can be used as an early screening tool to obtain pressure build-up estimations when detailed geological data are limited. These models cannot substitute detailed geological modeling in which geological features affect the flow and the results.

## 6. Conclusions

Modeling $CO_2$ and pressure plume propagation in the reservoir and around the wellbore is an important component for understanding viability and risks of CCUS. $CO_2$ injection can create pressure buildup near the wellbore and at the top of the reservoir which can hydraulically fracture the caprock or trigger leakage. Detailed geological characterization and numerical modeling are primary tools that can estimate the pressure buildup due to $CO_2$ injection. However, these models are expensive to develop. Here, we presented

proxy models for pressure buildup around the wellbore and average pressure buildup in the reservoir due to $CO_2$ and low-salinity water injection. Simple proxy models offer a quick way to screen the reservoir and eliminate the need to develop time-consuming geological models and numerical simulations for primary assessments.

Our study showed that when injecting $CO_2$, the pressure buildup was higher at the top of the aquifer compared with the buildup associated with injecting a similar mass of low-salinity water. We also found that permeability was the most important factor controlling pressure buildup around the wellbore during $CO_2$ injection, but average reservoir pressure buildup was affected primarily by porosity. Pressure buildup due to $CO_2$ injection is more than water injection (the same mass) because $CO_2$ is less dense and occupies more volume. Thus, injecting $CO_2$ is more likely to induce seismicity than injecting a similar mass of water. The average pressure buildup around the wellbore under the described conditions is in the order of 0.1 MPa. When injection ceases, this pressure buildup diffuses into the reservoir and is significantly reduced in the subsequent years. This study shows the value of proxy modeling in reducing the time and cost of initial screening candidate reservoirs for CCUS projects.

**Author Contributions:** Conceptualization, E.A. and E.H.; methodology, E.A.; resources, E.H.; writing—original draft preparation, E.A.; writing—review and editing, E.H. and F.H.; funding acquisition, E.H. All authors have read and agreed to the published version of the manuscript.

**Funding:** This work is funded by the Department of Energy under Award Number DE-FE0031623, Midcontinent Stacked Storage Hub. Neither the United States Government nor any agency thereof, nor any of their employees, makes any warranty, express or implied, or assumes any legal liability or responsibility for the accuracy, completeness, or usefulness of any information, apparatus, product, or process disclosed.

**Data Availability Statement:** The data presented in this study are available within the article. More detailed data are available at https://www.kgs.ku.edu/PRS/ICKan/index.html (accessed on 1 January 2023) and https://www.kgs.ku.edu/PRS/IMSCSH/index.html (accessed on 1 January 2023). Further inquiries can be directed to the corresponding author.

**Acknowledgments:** This work benefited from discussions with Jennifer Raney. Authors thank Julie Tollefson for preparing this manuscript for print.

**Conflicts of Interest:** The authors declare no conflict of interest.

## Abbreviations

| | |
|---|---|
| BHP | Bottom Hole Pressure (kPa) |
| $P_{res}$ | Average reservoir pressure (kPa) |
| $K$ | Permeability (mD) |
| $K_{VH}$ | Vertical to horizontal permeability ratio (unitless) |
| $\phi$ | Porosity (unitless) |
| $R_i$ | Injection rate ($m^3$/day) |
| $T$ | Temperature (°C) |

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
