# Peer review of "Simple Statistical Models for Predicting Overpressure Due to CO2 and Low-Salinity Waste-Fluid Injection into Deep Saline Formations"

_water, doi:10.3390/w15040648_

Round 1

Reviewer 1 Report

This manuscript is on the topic of establishing statistical models to predict overpressure due to  CO2 and low-salinity waste-fluid injection into deep saline formations. Although the technical approach and the execution of the computational work are sound, I have a few questions to the authors as detailed below:

1.      The major issue I had with this manuscript is the novelty of this study. The Introduction section elaborated of the research background in the first several sections. The last section in Introduction detailed the research objective and computational approaches. However, it seems that all the computational models have been introduced previously. Then, what is the novelty or originality of this study? Whether the authors introduced novel algorithm or any form of novel scholarly contributions to academia?

2.      Following the first comment, I can hardly find anywhere in this manuscript to articulate the novelty or significance of this piece of work. The authors attempted to elaborate the significance at the end of the Introduction and Conclusion sections. However, as detailed as above, the methodology of the work is not new with limited contributions to the academic understanding in this field. Thus, the elaboration of novelty of this work is quite weak. The authors have to improve the elaboration of the novelty of this article. Otherwise, this manuscript cannot be accepted.

3.      Whether the authors can comment on the applicability of this study: whether the conclusions of this study are only suitable for western Kansas reservoir or reservoir of a similar type of a wide range of reservoirs ?

Author Response

Thank you for the comment. 

  1. We agree the statistical methods already exist in the literature. The novelty of the work is in the presented data and the simple proxy models which can be used as an early screening tool to obtain pressure build-up estimations without detailed simulations. These models are especially useful when geological data on the reservoir are limited (e.g. deep saline aquifers across the US midcontinent).
  2. We added more information at the end of introduction. “The presented statistical models can be used as an early screening tool to screen large databases of reservoirs to select the most attractive ones for CO2 storage. These models are especially useful when geological data on the reservoir are limited (e.g. deep saline aquifers across the US midcontinent).”

  3.  

    We commented on this in the discussion section “These statistical models are relevant when the reservoir parameters are within the ranges of parameters given in Table 1”.

Reviewer 2 Report

1."Finally, the presented proxy models can be used as an early screening tool to obtain pressure build-up estimations when detailed geological data are limited. These models cannot substitute detailed geological modeling in which geological features affect the flow and the results." - there are the most important sentences from the paper.

2. Table of shortcut is necessary. There is no explanation of BHP in text

3. mD not md

4. More detailed description of parameters presented in Fig.9.

Author Response

Thank you for the comments.

  1. We inserted this sentence in the introduction as well. 
  2. We added a table for abbreviations (Table 2) to the manuscript. 
  3. We modified md to mD.
  4.  We included Table 1 for the list of parameters and put a reference under figure 9 to that table.

Round 2

Reviewer 1 Report

Accept in present form